# MicroRNAs as Potential Biomarkers of Post-Traumatic Epileptogenesis: A Systematic Review

**DOI:** 10.3390/ijms242015366

**Published:** 2023-10-19

**Authors:** Anastasia A. Vasilieva, Elena E. Timechko, Kristina D. Lysova, Anastasia I. Paramonova, Alexey M. Yakimov, Elena A. Kantimirova, Diana V. Dmitrenko

**Affiliations:** Department of Medical Genetics and Clinical Neurophysiology of Postgraduate Education, V.F. Voino-Yasenetsky Krasnoyarsk State Medical University, Krasnoyarsk 660022, Russia; drroptimusprime@gmail.com (A.A.V.); e.e.timechko@yandex.ru (E.E.T.); kris_995@mail.ru (K.D.L.); korolek_xd@bk.ru (A.I.P.);

**Keywords:** microRNA, biomarker, epilepsy, traumatic brain injury, ischemic stroke

## Abstract

Structural or post-traumatic epilepsy often develops after brain tissue damage caused by traumatic brain injury, stroke, infectious diseases of the brain, etc. Most often, between the initiating event and epilepsy, there is a period without seizures—a latent period. At this time, the process of restructuring of neural networks begins, leading to the formation of epileptiform activity, called epileptogenesis. The prediction of the development of the epileptogenic process is currently an urgent and difficult task. MicroRNAs are inexpensive and minimally invasive biomarkers of biological and pathological processes. The aim of this study is to evaluate the predictive ability of microRNAs to detect the risk of epileptogenesis. In this study, we conducted a systematic search on the MDPI, PubMed, ScienceDirect, and Web of Science platforms. We analyzed publications that studied the aberrant expression of circulating microRNAs in epilepsy, traumatic brain injury, and ischemic stroke in order to search for microRNAs—potential biomarkers for predicting epileptogenesis. Thus, 31 manuscripts examining biomarkers of epilepsy, 19 manuscripts examining biomarkers of traumatic brain injury, and 48 manuscripts examining biomarkers of ischemic stroke based on circulating miRNAs were analyzed. Three miRNAs were studied: miR-21, miR-181a, and miR-155. The findings showed that miR-21 and miR-155 are associated with cell proliferation and apoptosis, and miR-181a is associated with protein modifications. These miRNAs are not strictly specific, but they are involved in processes that may be indirectly associated with epileptogenesis. Also, these microRNAs may be of interest when they are studied in a cohort with each other and with other microRNAs. To further study the microRNA-based biomarkers of epileptogenesis, many factors must be taken into account: the time of sampling, the type of biological fluid, and other nuances. Currently, there is a need for more in-depth and prolonged studies of epileptogenesis.

## 1. Introduction

Epilepsy is a neurological disease characterized by a predisposition to the occurrence of epileptic seizures, as well as neurobiological changes and cognitive, psychological, and social impairments [1]. Epilepsy is one of the most common diseases of the nervous system. In the world, according to the WHO, there are about 75 million people diagnosed with epilepsy [2].

Structural or post-traumatic epilepsy (PTE) develops as a result of damage to brain tissue caused by traumatic brain injury (TBI), stroke, infectious diseases of the brain, etc. The frequency of brain damage in people of working age is significant, so the problem of predicting the development of post-traumatic epilepsy is an urgent task for researchers [3,4].

The types of post-traumatic seizures are classified as follows: (1)Immediate—arising in the first 24 h after a traumatic event;(2)Early—from 24 h to 1 week after the traumatic event;(3)Late—occurring more than 1 week after the traumatic event [3].

At the same time, immediate and early seizures are not considered truly “epileptic”, since it is assumed that they are provoked by an initiating event, and are not a consequence of the molecular, biochemical, and histological rearrangements of brain tissue [4]. The latent period of post-traumatic epileptogenesis is characterized by the absence of epileptic seizures from the moment of injury to the first late seizure. This period can last up to 10 years or more [5].

Epileptogenesis is a pathological process of the reorganization of tissues and cells in the brain, leading to the formation of hyperexcitable neural networks and the subsequent appearance of epileptiform activity. It was previously believed that epileptogenesis occurs only in the latent period—between the epileptogenic event and the onset of the first clinical seizure. Currently, according to studies [6,7,8], epileptogenesis is a longer process and includes the mechanisms of disease progression, since the frequency and severity of seizures increase with time. Pathological mechanisms leading to the development of epileptogenesis and subsequent chronicization of structural epilepsy are considered to be neuroinflammation of the brain (production of specific cytokines and chemokines that activate glia, which can lead to the recruitment of macrophages and lymphocytes to a dysfunctional BBB, leading to chronic inflammation and neurodegeneration), dysfunction of the blood–brain barrier, oxidative stress (proliferation of reactive oxygen species capable of organic damage to neuronal cells), apoptosis of neurons, activation of astrocytes, dendritic growth, gliosis (increased number, altered morphology with hypertrophy of the soma and processes, spatial overlap and various functional changes in astrocytes, leading to reorganization of the neuronal network and the formation of hyperexcitable areas), etc. [4,5].

The problem of assessing the risks of developing post-traumatic epilepsy after traumatic brain injury and stroke in the early stages and during the progression of the disease is one of the most urgent topics in the study of epilepsy. Currently, it is impossible to predict the development of the epileptogenic process after the initiating event (TBI, stroke, etc.); it is only possible to diagnose if the disease already exists [4].

The standard methods for diagnosing epilepsy today are electroencephalography (EEG) and neuroimaging methods such as computed tomography (CT) and magnetic resonance imaging (MRI) [9].

The use of non-invasive and minimally invasive biomarkers for diagnosing and predicting the development of epilepsy can serve as an alternative to classical diagnostic methods. A biomarker is an indicator of biological and physiological processes occurring in the body, as well as pathogenic processes or reactions to various influences, including therapeutic intervention [10]. According to the BEST (Biomarkers, EndpointS, and other Tools) Resource, there are the following types of biomarkers: susceptibility (risk) biomarkers; diagnostic biomarkers; biomarkers for disease monitoring; prognostic biomarkers; response biomarkers; and biomarkers of safety/toxicity [10].

A potential prognostic biomarker of epileptogenesis is microRNAs (miRNAs), endogenous non-coding RNA molecules that can bind to one or more mRNAs, mainly inducing their degradation and, consequently, repression of translation [11]. MicroRNA processing and the mechanism of action are summarized in Figure 1. 

The advantages of using microRNAs are the minimally invasive methods for determination, relative stability in biological fluids, and the simplicity and low cost of the methods used [12].

This review analyzed manuscripts related to biomarkers for epilepsy, TBI, and ischemic stroke. The purpose of the review was to analyze the available information and search for miRNAs that have potential as biomarkers and predictors of epileptogenesis.

## 2. Methods

When writing the systematic review, the following aggregators of scientific journals were used: MDPI, PubMed, ScienceDirect, and Web of Science. Scientific articles were searched using the following keywords: microRNA, biomarker, epilepsy, traumatic brain injury, ischemic stroke. 

Scientific articles from 2010 to 2023 were analyzed, including original research and review articles. Publications were included in the systematic review if they were published in English and complied with the inclusion criteria defined below. Duplicates were excluded from the review. Research publications containing information on aberrant miRNA expression in plasma/serum/saliva/cerebrospinal fluid in epilepsy, TBI, and ischemic stroke in humans were included.

Only studies with statistically significant aberrant miRNA expression (according to AUC criteria and/or *p*-value, depending on the criteria used in the original research) were analyzed. The review excluded manuscripts that investigated miRNA expression in animal models and in brain tissues.

This review was produced in accordance with the Preferred Reporting Items for Systematic Reviews and Meta-Analyses (PRISMA 2020). The block diagram is shown in Figure 2.

### 2.1. Inclusion Criteria

#### Study Material

The material included studies of miRNA expression in patients with epilepsy, traumatic brain injury, and ischemic stroke. No limitations were placed on age, ethnicity, country of origin, family history of recurrent miscarriage, or any other patient demographics. Only original studies were included in the review.

The material on patients’ investigation samples included cerebrospinal fluid, plasma, and blood.

### 2.2. Exclusion Criteria

The studies of patients with epilepsy were limited by type of epilepsy. Genetic forms of epilepsy were excluded. The study of brain tissue was also excluded from the analysis. Animal model studies were also excluded from the review. Reviews were also excluded.

## 3. Results and Discussion

The search for biomarkers of post-traumatic epilepsy is difficult. Epileptogenesis is a long, non-specific, and dynamic process. It may be necessary to investigate different panels of biomarkers depending on the stage and etiology of the initiating event. Another problem is the search for those biomarkers that are specific for epileptogenesis, and not the severity of brain damage.

When searching for biomarkers of epileptogenesis, a number of factors must be taken into account. MicroRNA expression profiles may differ depending on the type of biological fluid taken for analysis. There is evidence [13] that cerebrospinal fluid is more informative than peripheral blood. However, the sampling procedure is more invasive, so CSF is less frequently studied. It is worth paying attention to the study of exosomal miRNAs, which is independent of the total content in the biofluid. Exosomal miRNAs can also be more informative [14]. Despite this, most studies are aimed specifically at the general study of the miRNAs profile in peripheral blood due to the greater availability of the method.

The timing of biosampling also plays an important role in miRNA analysis since miRNA expression in the acute phase of an epileptic seizure/trauma/stroke and miRNA expression, which indicates molecular and metabolic changes caused by trauma, may differ. Only a few publications that have studied models of epilepsy examine the biomaterial immediately after a seizure [14,15]. Obviously, this is difficult in studying the model of epilepsy. In TBI and ischemic stroke, almost all studies indicate the time of biomaterial collection.

### 3.1. Circulating miRNAs in Epilepsy

The development of epilepsy occurs due to a series of pathological changes that form a repetitive cycle of excitation in the hippocampus, leading to the appearance of seizures. In patients with epilepsy, many molecular anomalies in various metabolic processes are observed in the brain [16]. It has been proven that they are associated with the impaired transcriptional and post-transcriptional regulation of genes in epileptogenesis [17].

Thirty-one manuscripts were analyzed that examined circulating miRNAs for which aberrant expression was observed in patients with epilepsy (Table 1). miRNAs, as biomarkers of epilepsy, have been extensively studied in various publications, but the association with post-traumatic epilepsy is not often made. Chun-Hong Shen et al., in their study [18], determined that miR-145-5p was reduced in patients with drug-resistant epilepsy compared to healthy volunteers. The decreased expression of these miRNAs correlated with a history of TBI or encephalitis in patients. However, in two other studies, miR-145 was overexpressed in patients with mesial temporal lobe epilepsy immediately after a seizure [15] and in patients with established hippocampal sclerosis [19].

Most often, blood plasma or serum was studied, but cerebrospinal fluid was studied in some publications [20,21]. In some cases, miRNA expression was compared in various biofluids. Ye Zhao et al. found a significant increase in miR-1275 expression in the serum of patients with epilepsy of unknown etiology, but it was reduced in the CSF [21]. Similarly, Rana Raoof et al. did not find a statistically significant correlation between miR-19b-3p, miR-21-5p, or miR-451a expression in CSF compared with plasma [22]. Therefore, it should be assumed that the information content of studies of different types of biological fluids will vary.

It is worth paying attention to studies of epilepsy in pediatric patients. It is generally accepted that epileptogenic mechanisms in children differ from those in adults [23]. However, data on the microRNA–age correlation are insufficient. Some miRNAs show similar expression patterns in both children and adults and may not be related to age. Examples include miR-106b [24,25,26] and miR-155 [27,28,29].

Although tissue expression studies were excluded from the review, some authors performed a comparative analysis between tissue and circulating miRNAs. One study found that the miR-145, miR-181c, miR199a, and miR-1183 miRNAs circulating in biological fluids were overexpressed in patients with mesial temporal lobe epilepsy, but hypoexpressed in hippocampal tissues [19]. In contrast, Yuqiang Sun et al. did not find a difference in miR-129-2-3p expression levels in plasma and cerebral cortex tissues in patients with drug-resistant epilepsy—in both cases, it was hyperexpressed.

**Table 1 ijms-24-15366-t001:** Circulating miRNAs in epilepsy.

Experimental Design	Sample	Sampling Time	MiRNAs	Significance		Year	Link
				AUC	*p*-Value		
TLE/HV	CSF		miR-219↓		<0.001	2014	[20]
DR E/HV	Serum		miR-194-5p↑ miR-301a-3p↑ miR-30b-5p↑ miR-342-5p↑ miR-4446-3p↑	0.740 0.893 0.684 0.721 0.703		2015	[30]
DR E/HV	Serum		miR-4521↑	0.718		2016	[31]
DR E/HV	Plasma		miR-153↓		0.001	2016	[32]
DR E/HV	Plasma, brain tissue		miR-129-2-3p↑	0.778	0.0008		[33]
E/HV	Serum		miR-106b↑ miR-146a↑ miR-301a↑ miR-194-5p↓	0.786 0.774 0.686 0.696		2016	[24]
E/HV	Serum		miR-378↑ miR-30a↑ (associated with the frequency of seizures) miR-106b↑ miR-15a↑		<0.010	2016	[25]
Seizure mTLE/mTLE	Serum	30 min after seizure	miR-143↑ miR-145↑ miR-532↑ miR-365a↑		<0.050	2016	[15]
DR E/HV	Plasma		miR-323a-5p↑		0.014	2017	[34]
mTLE+HS/HV	Plasma		miR-3613-5p↑ miR-4668-5p↓ miR-8071↓ miR-197-5p↓ miR-4322↓ miR-6781-5p↓	0.844 0.789 0.932 0.802 0.714 0.781		2017	[35]
DR-mTLE/mTLE/HV	Plasma		miR-134↓	0.671		2017	[36]
TLE/SE/HV	CSF		TLE/SE: miR-19b-3p↑ miR-21-5p↑ miR-204-5p↑ miR-223-3p↑ miR-451a↑ miR-886- 3p↓ TLE/HV: miR-19b-3p↑ miR-21-5p↑ miR-204-5p↑ miR-223-3p↑ miR-451a↑ miR-886-3p↑	0.780 0.800 0.910 0.730 0.830 0.800		2017	[22]
E/HV	Plasma		miR-153↓		<0.010	2018	[37]
E/HV	Serum		miR-155↑		<0.010	2018	[27]
E/E after seizure/HV	Plasma	24 h	TLE/TLE after seizure: miR-27a-3p↓ miR-328-3p↑ miR-335-5p↓ TLE/HV: miR-27a-3p↓ miR-328-3p↑ miR-335-5p↓		<0.010 <0.050 <0.050 <0.010 <0.010 <0.010	2018	[14]
DR E/mTLE/HV	Plasma		DR E/HV: miR-145-5p↓ mTLE/HV: miR-145-5p↓	0.632 0.829		2019	[18]
mTLE+HS/HV	Serum		miR-145↑ miR-181c↑ miR199a↑ miR-1183↑		0.005 0.030 0.010 0.001	2019	[19]
DR E/E	Serum	Seizures during 12 months/Without seizures during 12 months	miR-146a-5p↑ miR-134-5p↑	0.640 0.617	<0.001	2020	[38]
ChE/HV	Plasma		miR-106b↑ miR-146a↑	0.885 0.763		2019	[26]
GGE/HV	Serum		miR-146a↑ miR-155↑ miR-132↓	Average: 0.850		2020	[28]
IE/HV	Plasma		miR-125a↓ miR-181a↓	0.815 0.704	0.001 0.001	2020	[39]
mTLE+HS(Engel I, Engel III-IV)/HV	Serum		E I/HV: miR-328-3p↑ miR-654-3p↑ E III-IV /HV: miR-328-3p↑ miR-654-3p↑ E I/E III-IV: miR-328-3p↑ miR-654-3p↑		0.001 0.004 <0.001 0.89 <0.001 0.190		[40]
UEE/HV	Serum, CSF		Serum: miR-1275↑ CSF: miR-1275↓			2020	[21]
TLE/HV	Serum		miR-142↑ miR-146a↑ miR-223↑		<0.001 0.020 <0.001	2021	[41]
Ch TLE/HV	Plasma		miR-194-5p↓	0.896		2021	[42]
TLE/HV	Plasma		miR-146a↑ miR-132↑	0.808 0.791		2022	[43]
ChE/HV	Plasma (sEVs)		miR-584-5p↑ miR-342-5p↑ miR-150-5p↑	0.846 0.835 0.826		2022	[44]
ChE/HV	Serum		miR-155	0.813		2022	[29]
ChE/HV	Serum		miR-324-5p↑ miR-146a-5p↓ miR-138-5p↓ miR-187-3p↓			2022	[45]
E/HV	Serum		miR-378↓ miR-575↓		<0.001 <0.001	2022	[46]
mTLE+HS/HV	Serum		miR-629-3p↑ miR-1202↑ miR-1225-5p↑			2022	[47]

TLE—temporal lobe epilepsy; HV—healthy volunteers; CSF—cerebrospinal fluid; DR—drug-resistant epilepsy; mTLE—mesial temporal lobe epilepsy; HS—hippocampal sclerosis; SE—status epilepticus; E—epilepsy; GGE—genetic generalized epilepsy; ChE—childhood epilepsy; UEE—unknown etiology epilepsy; IE—idiopathic epilepsy; Engel I—good surgical outcome; Engel III—bad surgical outcome; ↑—upregulation; ↓—downregulation; AUC—area under the curve.

### 3.2. Circulating miRNAs inTraumatic Brain Injury

The risk of developing post-traumatic epilepsy after TBI is quite high. In a study conducted by Meral A Tubi et al., 45.7% of a cohort of patients with TBI of varying severity acquired PTE within two years of injury [48].

The progression of injury towards epileptogenesis occurs in a state of secondary injury—a few hours or days after it. At this time, a number of complex molecular and structural transformations occur, leading to the development of epilepsy in certain circumstances [3]. Structural changes include neurodegeneration [49], gliosis [50], damage or sprouting of axons [51], dendritic plasticity [52], damage to the blood–brain barrier [53], and recruitment of inflammatory cells in brain tissue [54].

To diagnose and predict the outcome of TBI, the Glasgow scale is used, based on eye-tracking and the verbal and motor activity of the patient. Together with the presence and amount of loss of consciousness, the severity of the injury is determined [55]. This method, in combination with classical CT and MRI, does not always make it possible to identify the severity and focus of the injury [4]. It is also difficult to find the prerequisites for the development of epileptogenesis [4].

Table 2 summarizes the aberrantly expressed miRNAs in TBI patients. Studies of circulating microRNAs as biomarkers of TBI often aim to diagnose TBI after a head injury, as well as to determine the severity. For example, Manish Bhomia et al. found the aberrant expression of 18 microRNAs in moderate TBI and 20 in severe TBI. Subsequently, verification analyses of 10 miRNAs were carried out in the cerebrospinal fluid of patients with various TBIs. A correlation was also made between CT-positive and CT-negative patients. It was found that miR-328, miR-362, and miR-486 are potential biomarkers of severe versus moderate TBI. The expression of selected miRNAs differed in CT-negative patients—some of them were diagnosed with TBI, but the authors did not identify a separate cohort [56]. It is also worth paying attention to prolonged studies in which miRNA levels are measured over a long period of time [57,58,59,60].

MicroRNAs can be predictors of comorbidities even years after the initiating event. In one study, the cohort of patients included individuals with TBI-induced hypopituitarism [58]. In patients who developed hypopituitarism after TBI, there was an increase in miR-3610 by 5.0 times on the 1st day, 8.5 times on the 7th day, 2.3 times on the 28th day, and 10.0 times at year 5 compared to healthy controls. Another miRNA, miR126-3p, was reduced 11-fold on the 1st day and 12-fold on the 28th day, while at the 5th year after trauma, there was a 12-fold increase compared to the control group. In addition, it was noted that miR-126-3p showed a 15-fold increase at year 5 after trauma when patients with and without hypopituitarism after TBI were compared. The problem with such studies is that there are difficulties in the long-term monitoring of patients. Therefore, such studies involve patients with already-established diagnoses and a history of TBI of a certain prescription. Nevertheless, such a research model is optimal in the study of PTE.

**Table 2 ijms-24-15366-t002:** Circulating miRNAs in TBI.

Experimental Design	Sample	Sampling Time	MiRNAs	AUC	*p*-Value	Year	Link
sTBI/HV	Plasma	24 h after trauma	miR-16↓ miR-92a↓ miR-765↑	0.890 0.820 0.860		2010	[61]
sTBI/OT			miR-16↓ miR-92a↓ miR-765↑	0.820 0.830 0.790			
sTBI/HV			miR-16↓ miR-26a↑ miR-92a↓ miR-638↑ miR-765↑	0.820 0.730 0.780 0.510 0.680			
TBI of varying severity/HV	Serum	24 h after trauma–21 days after trauma	miR-93↑ miR-191↑ miR-499↑	1.000 0.727 0.801		2016	[57]
sTBI (coma)/HV	CSF	Coma during two weeks after trauma	miR-141↑ miR-572↑ miR-181a-star↑ miR-27b-star↑ miR-483-5p↑ miR-30b↑ miR-431-star↑ miR-193b-star↑ miR-499-3p↑ miR-1297↓ miR-33b↓ miR-933↓ miR-449b↓		0.005 0.002 0.001 0.012 0.003 0.030 0.006 0.021 0.002 0.013 0.016 0.009 0.042	2016	[62]
TBI/HV	Serum	24 h after trauma	miR-155↓	0.726		2019	[63]
TBI of varying severit/OT/HV	Serum	24–48 h after trauma	miR-151-5p↑ miR-195↑ miR-20a↑ miR-328↑ miR362-3p↑ miR-30d↑ miR-451↑ miR-486↑ miR-505*↑ miR-92a↑	0.660 0.810 0.780 0.730 0.790 0.750 0.820 0.810 0.820 0.860		2016	[56]
TBI+ hypopituitarism /HV	Serum	1, 7, 28 days after trauma	miR-3907↑ miR126-3p↓		< 0.050	2016	[58]
лЧМT/ЗД	Serum	24 h–15 days after trauma	miR-425-5p↓ miR-21↑ miR-502↓ miR-335↑	1.000 0.961 1.000 0.990		2017	[59]
TBI of varying severity/HV	Saliva, CSF		miR-182-5p↓ miR-221-3p↓ mir-26b-5p↓ miR-320c↓ miR-29c-3p↑ miR-30e-5p↑	Average 0.852		2018	[64]
TBI of varying severity/HV	Plasma		miR-6867-5p ↑ miR-3665↑ miR-328-5p↑ miR-762↑ miR-3195↑ miR-4669↑ miR-2861↑	0.854 0.877 0.888 0.916 0.899 0.907 0.913		2018	[65]
mTBI/HV	Saliva, Serum		miR-10b-5p↑ miR-30b-5p↑ miR-3678-3p↓ miR-455-5p↓ miR-5694↓ miR-6809-3p↓ miR-92a-3p↓	Average 0.890		2019	[66]
sTBI/HV	Serum	<24 h after trauma	miR-103a-3p↑ miR-219a-5p↑ miR-302d-3p↑ miR-422a↑ miR-518f-3p↑ miR-520d-3p↑ miR-627↑		<0.050	2019	[67]
mTBI/HV	Serum	24-48 h after trauma	miR-151-5p↑ miR-362-3p↑ miR-486↑ miR-505↑ miR-499↑ miR-625↑ miR-638↑ miR-381↑		<0.050	2020	[68]
TBI/HV	Serum	<24 h after trauma	miR-124-3p↑ miR-219a-5p↑ miR-9-5p↑ miR-9-3p↑ miR-137↑ miR-128-3p↑	0.710 0.890 0.890 0.780 0.810 0.740		2020	[69]
Isolated TBI/HV	Plasma/Serum	6 h after trauma	miR-423-3p↑	0.790		2020	[70]
TBI/HV	Serum	<24 h after trauma	miR-93↑ miR-191↑	0.712 0.660		2020	[71]
TBI/HV	Plasma	<24 h after trauma	miR-203b-5p↓ miR-203a-3p↓ miR-206↑ miR-185-5p↑	0.840		2020	[72]
modTBI/HV	Plasma	less than 1 day, 7, 28 days after trauma	miR-32-5p↑	0.700–0.900		2022	[60]

sTBI—severe traumatic brain injury; HV—healthy volunteers; CSF—cerebrospinal fluid; OT—orthopedic injury; mTBI—mild traumatic brain injury; modTBI—moderate craniocerebral injury; ↑—upregulation; ↓—downregulation; AUC—area under the curve.

### 3.3. Circulating miRNAs in Ischemic Stroke

Ischemic stroke, along with TBI, is one of the main causes of post-traumatic epilepsy [73]. The incidence of PTE ranges from 3% to 25% depending on various factors [74]. Neurological deficit, disability, and damage to the cerebral cortex and hippocampus are the main factors in the development of post-stroke seizures [75]. Seizures with early onset do not cause epilepsy, but are considered a risk factor. However, late seizures occurring more than a week later pose a greater risk of PTE. This is due to structural changes in the neural networks of the brain, such as dysfunction of the blood–brain barrier. Late attack causes multiple mechanisms—gliotic scarring with changes in membrane properties, albumin extravasation, changes in astrocytic proteins, and brain irritation due to hemosiderin deposits, as well as thrombin stimulation of the protease-activated receptor [76]. The pathogenesis of PTE in terms of molecular markers is associated with elevated plasma interleukin-6 (IL-6), high mobility group B1 protein, and low-serum neuropeptide Y [76]. The dysregulation of miRNA expression can also be a factor regulating the processes described above. It was previously reported [77] that miRNAs are involved in the processes of oxidative stress, inflammation, and apoptosis; in this case, either protective or pathogenic functions are manifested [77].

The publications that have studied miRNA expression in ischemic stroke are listed in Table 3. The clinical use of miRNAs as biomarkers of ischemic stroke is most often in diagnostics—in determining differences between controls and a group of patients. However, other goals may be pursued, such as establishing the etiology of a stroke. The publication of G. Long et al. [78] included patients with ischemic stroke of various etiologies: damage to small vessels, cardioembolism, and atherosclerosis of large vessels. It was found that the levels of let-7b miRNA were different in patients with atherosclerosis of large vessels and were reduced compared to the control. In other patients, the levels of this miRNA were significantly higher than in controls.

One of the goals in the study of miRNAs in ischemia is to establish a correlation with possible deterioration in the patient’s condition. The authors of one of the studies showed that miR-132 miRNA levels can be statistically correlated depending on the presence of post-stroke cognitive impairment in patients [79]. Hyperexpressed levels of miR-124-3p positively correlated with mortality within 3 months in patients with ischemic stroke [80].

**Table 3 ijms-24-15366-t003:** Circulating miRNAs in ischemic stroke.

Experimental Design	Sample	Sampling Time	MiRNAs	AUC	*p*-Value	Year	Link
IS/HV	Serum	3, 4, 7 days after attack	miR-210↓		0.001	2011	[81]
IS/A/HV	Serum		miR-21↑ miR-221↓		<0.0001 0.0002	2013	[82]
IS/HV	Serum	24 h, 1 week, 4 weeks, 24 weeks and 48 weeks after attack.	24 h–24 weeks: miR-30a↓ miR-126↓ let-7b↑ (but let-7b ↓ for atherosclerosis of large vessels)	0.910–0.930 0.920–0.940 0.920–0.930		2013	[78]
IS/HV	Serum		miR-145↑			2014	[83]
IS/HV	Serum	72 h after attack	miR-223↑		<0.050	2014	[84]
IS/HV with vascular risk factors	Serum	>2 weeks after attack	miR-122↓ miR-148a↓ let-7i↓ miR-19a↓ miR-320d↓ miR-4429↓ miR-363↑ miR-487b↑		0.047 0.009 0.023 0.030 0.020 0.034 0.037 0.044	2014	[85]
ACI/HV	Plasma		miR-21↑ miR-24↓		<0.050	2014	[86]
AS/HV	Plasma	0–24 h after attack	hsa-miR106b-5p↑ hsa-miR-4306↑ hsa-miR-320e↓ hsa-miR320d↓	0.962 0.877 0.981 0.987		2014	[87]
IS/HS	Plasma		miR-124-3p↓ miR-16↑		0.010 0.039	2014	[88]
AcuteIS/HV	Serum	24 h after attack	miR-124↓ miR-9↓		0.011 <0.001	2015	[89]
IS/HV	Serum	24 h after attack	miR-32-3p↑ miR-106-5p↑ miR-532-5p↓		<0.050 <0.010 <0.010	2015	[90]
AS/IS/HV	Serum	24 h after attack	miR-145↑ miR-23a↓ miR-221↓	0.794 0.816 0.819	<0.001 <0.001 <0.001	2015	[91]
IS/HV	Serum		miR-15a↑ miR-16↑ miR-17-5p↑	0.698 0.820 0.784		2015	[92]
IS/HV	Serum, CSF		let-7e↑		0.033	2015	[93]
IS/HV	Serum		miR-9,-22,-23,-27,-125,-524↑ miR-9,-30,-33,-124,-135, -181,-197,-218,-330,-452↓		<0.100 <0.150	2015	[94]
IS/HV	Serum		miR-107↑ miR-128b↑ miR-153↑	0.970 0.903 0.893		2016	[95]
PSCI/PSCN/HV	Serum	-	miR-132↑	0.961		2016	[79]
Fatal post-stroke/survivors	Plasma	<24 h after attack	miR-124-3p↑ miR-16↓	0.750 0.690		2016	[80]
IS/HV	Serum	<72 h after attack	let-7i↓		<0.00001	2016	[96]
IS/HV TIA/HV	Plasma	<24 h after attack	miR-125a-5p↑ miR-125b-5p↑ miR-143-3p↑		IS/HV 1.1 × 10^−7^ 8.2 × 10^−9^, 2.4 × 10^−9^ TIS/HV 0.003 0.003 0.005	2017	[97]
IS/HV	Plasma	1–3 days after attack 4–14 days after attack	miR-422a↑ miR-125b-2-3p↓ miR-422a↓ miR-125b-2-3p↓	0.769 0.971 0.889		2017	[98]
IS/HV	Serum	<24 h after attack	miR-221-3p↓ miR-382-5p↓	0.8106 0.7483		2017	[99]
IS/TIA/HV	Serum	<24 h after attack	IS: miR-23b-3p↑ miR-29b-3p↑ miR-181a-5p↑ miR-21-5p↑ TIS: miR-23b-3p↑ miR-29b-3p↑ miR-181a-5p↑		<0.001 <0.001 0.027 <0.001 <0.001 0.001 0.007	2017	[100]
IS/HV	Serum	1–3 days after attack	miR-146a↓	0.910		2017	[101]
IS/HV	CSF	<48 h after attack	miR-9-5p↑ miR-128-3p↑		<0.050	2017	[102]
IS/HV	Serum	<48 h after attack	miR-4656,-432, -503↑ miR-874↓		<0.050	2018	[103]
IS/HV	Serum	<24 h after attack	miR-146b↑		0.863	2018	[104]
IS/HV	Serum	<72 h after attack	miR-15a↑		<0.050	2018	[105]
IS/HV	Serum	<24 h after attack	miR-148b-3p↓ miR-151b↑ miR-27b-3p↑	0.665 0.685 0.666		2018	[106]
IS/HV	Serum	<24 h after attack	miR-186-5p↓ miR-19a-3p↓ miR-32-5p↓ miR-340-5p↓ miR-579-3p↓ let-7e↑ miR-362-3p↑ miR-1238-5p↓		<0.050	2019	[107]
Acute IS/HV	Serum	<24 h after attack	miR-124↓		<0.001	2019	[108]
IS/HV	Serum	<24 h after attack	miR-146a↓		<0.001	2019	[109]
IS/HV	Serum		miR-451↑	0.912		2019	[110]
Acute IS/HV	Serum	<24 h after attack	miR-124↑		0.007	2020	[111]
IS/HV	Serum		miR-24↓ miR-29b↓	0.802 0.835		2020	[112]
IS/HV	Serum	<24 h 14 days after attack	miR-135b↑ (24 h after attack) miR-135b↑ (14 days after attack)	0.780		2020	[113]
IS/HV	Serum	<24 h after attack	miR-602↓	0.817			[114]
IS/HV	Plasma		miR-155↑	0.851		2020	[115]
IS/HV	Serum		miR-128↑		<0.001	2020	[116]
IS/HV	Serum	<72 h after attack	miR-124↓	0.953		2021	[117]
IS/HV	Serum	<72 h after attack	miR-148a↑ miR-342-3p↑ miR-19a↑ miR-320d↑	0.872 0.844 0.721 0.673		2021	[118]
Acute IS/HV	Serum		miR-9-5p↑ miR-128-3p↑	0.946 0.929		2021	[119]
IS/HV	Serum		miR-659-5p↑ miR-151a-3p↑ miR-29c-5p↑		0.003 0.0007 0.0007	2022	[120]
IS/A/HV	Plasma	IS <6 h after attack	miR-129-1-3p↑ miR-4312↑ miR-5196-3p(IS↑, A↓)		<0.001	2022	[121]
IS/HV	Serum	<24 h after attack	miR-451a↑ miR-574-5p↓ miR-142-3p↓	0.970		2022	[122]
IS/HV	Serum		miR-15b-5p↑ miR-184↑ miR-16-5p↑	0.581 0.607 0.719		2022	[123]
IS/HV	Serum		miR-6089↓		<0.050	2023	[124]

IS—ischemic stroke; HV—healthy volunteers; A—atherosclerosis; TIA—transient ischemic attack; CI—cerebral ischemia; ACI—acute cerebral infarction; AS—acute stroke; HS—hemorrhagic stroke; PSCI—post-stroke cognitive impairment; PSCN—post-stroke cognitive normality; ↑—upregulation; ↓—downregulation; AUC—area under the curve.

### 3.4. Discussions

According to the analyzed information, three microRNAs were found that exhibit aberrant expression in TBI, stroke, and epilepsy: miR-21 [22,59,82,86,100], miR-181a [39,62,100], and miR-155 [27,28,29,63,115]. A summary of the data is shown in Table 4 and also in Figure 3.

#### 3.4.1. miR-21

miR-21 microRNA is a well-known and well-studied biomarker of many diseases [125], is involved in inflammatory [126] processes, and is associated with cancer [127] and cardiovascular [128] diseases. The targets of this miRNA and related processes were analyzed using the MiRTarBase database and ShinyGO v.0.77 and TissueAtlas software (https://ccb-web.cs.uni-saarland.de/tissueatlas/ (accessed on 27 September 2023)). Figure 4 shows the biological processes regulated by miR-21. These processes are not specific to epilepsy. However, the processes of cell development, proliferation, and apoptosis remain important for such neurological disorders. miR-21-3p is most actively expressed in the arachnoid mater (Figure 5), which may indicate an association with epileptogenesis. Previously, only miR-21-5p was studied in the model of epilepsy [22].

A study by R. Raoof et al. [22] used miR-21-5p, miR-451a, and miR-19b-3p expression in epilepsy as a test material for CSF samples from patients with temporal lobe epilepsy, status epilepticus, and the control group. Patients suffering from other neurological diseases (Alzheimer’s disease, multiple sclerosis) were also included in order to test the specificity of biomarkers. The miR-21-5p expression was significantly higher in patients with status epilepticus than in patients with temporal lobe epilepsy and controls [22]. Also, the values were overestimated in patients with other neurological diseases.

The miR-21 expression in patients with TBI [59] of various severity (moderate, severe), as well as in patients with extra-cranial injury without TBI and in healthy controls, was measured in blood serum at different time points (0–1 h, 12 h, 48–72 h, 15 days) from injury. It was found that the miR-21 expression level is strongly overestimated in patients with severe TBI in comparison with other groups of patients during the entire observation period. After 6 months, miR-21 showed no predictive value.

In a model of ischemic stroke, the level of miR-21 was compared in serum in patients with stroke, atherosclerosis, and in the control group [82]. Serum levels of miR-21 were elevated in patients with atherosclerosis along with IS. After adjusting for traditional risk factors, miR-21 levels remained a significant predictor of stroke [82]. In another study [86], patients with acute ischemic infarction served as an experimental group; the expression data are similar to previous studies.

As shown earlier, miR-21 is not specific. The studies listed above show that miR-21 is overexpressed in the brain lesions of various etiologies. However, this miRNA may be useful for epileptogenesis in terms of controlling the neurodegenerative regulation associated with inflammation and apoptosis.

#### 3.4.2. miR-181a

miR-181a is a member of the miR-181 family, which consists of four microRNAs: miR181a, miR181b, miR181c, and miR181d. The impaired expression of these miRNAs is associated with the immune response, particularly with the differentiation of B- and T-lymphocytes [129]. miR-181a targets include heat shock protein GRP78 [130], transcription factor CREB1 [131], and the Bcl-2 family of proteins [132]. This microRNA is largely associated with protein modifications in the body (Figure 6), which may be of great interest for epileptogenesis. It was also found that miR-181a is predominantly expressed in the brain and spinal cord (Figure 7).

Publication [39] determined the relationship between miR-125a and miR-181a expression levels, IFN-γ and TNF-α production, and epileptogenic processes. The study involved children with an idiopathic form of epilepsy. The microRNA levels were significantly reduced in epileptic patients compared to healthy controls. However, the levels of pro-inflammatory cytokines in patients were slightly increased, and the authors indicated the absence of a correlation with miRNA expression. Based on this, it can be concluded that miR-181a is not associated with idiopathic epilepsy. It is likely that structural damage is necessary for its aberrant expression.

Another study [62] studied the expression of a spectrum of miRNAs, including miR-181a, in patients with severe TBI with coma. The sample for the study was CSF. The miR-181a expression level was 2.47 times higher in patients compared to healthy controls. In an IS model, miR-181a showed increased expression in a model of ischemic stroke and transient ischemic attack [100].

It is not known how specific miR-181a may be for epileptogenesis, but there are a number of prerequisites for this. This miRNA is of interest from the point of view of metabolic functions. Studies also confirm its importance, but they are too few for accurate judgments.

#### 3.4.3. miR-155

miR-155 plays an important role in inflammatory processes, apoptosis, and cell proliferation [133,134]. An analysis of biological processes (Figure 8) showed that miR-155 is predominantly involved in cell migration and proliferation. The tissue expression of miR-155 is not specific; however, data are only available for miR-155-5p (Figure 9).

In epilepsy, the increased expression of miR-155 was observed in three independent studies [27,28,29]. Wei Duan et al. [27] found an overexpression of miR-155 in the blood serum of patients with temporal lobe epilepsy. It was concluded that miR-155 is associated with the PI3K/Akt/mTOR signaling pathway, inducing neuronal apoptosis. A study by R. Martins-Ferreira et al. [28] determined the expression patterns of the miR-146a, miR-155, and miR-132 microRNA panel in patients with genetic generalized epilepsy, all of which were overestimated. Exosomal miR-155 was also elevated in children with epilepsy in a study by Ya Liu et al. [29]. However, children with PTE were excluded from the cohort.

In patients with TBI, the level of miR-155 in the peripheral blood was reduced [63]. The authors associated the reduced expression of this microRNA with the formation of damage to the intestinal mucosa and further dysfunction by affecting the expression of the claudin1 protein.

In the model of acute and subacute ischemic stroke, the vesicular microRNA miR-155 was overestimated compared to the control by 1.6 and 2.5 times, respectively [115]. Higher levels were found in patients with mild to moderate stroke.

The common gene targets for these three microRNAs are BCL2, EGFR, and SOX6. According to the STRING database, the most significant pathways in which these genes are involved are intrinsic apoptotic signaling, EGFR signaling, and glial cell fate commitment with an indicated strength above 2.

#### 3.4.4. Other Potential miRNA-Based Markers

The microRNAs associated with TBI/epilepsy as well as IS/epilepsy are listed below and are shown in Table 4.

The most important potential biomarkers are miRNAs, whose expression is associated with activity from ion channels. Among them are miR-30b [135], miR-219a [136], miR-335 [137], miR-342 [138], and miR-30a [25].

miR-328-3p is widely studied in relation to cancer [139]. Despite this, it has shown itself as a biomarker for epilepsy [14,40] and for determining the severity of TBI [56]. Also, miR-30a [140,141], miR-15a [142], and miR-125a [143] are associated with cancer.

miR-153 is associated with refractory epilepsy in terms of its effect on the hypoxia factor HIF-1α [37].

miR-106b, as an immune response factor [26], is an interesting target for epileptogenesis studies. It has been actively studied both in the model of epilepsy [24,25,26] and in ischemic stroke [87,90]. Similarly, miR-146a is one of the most frequently studied epilepsy and IS biomarkers, as shown in the respective tables. Despite this, this miRNA in ischemic stroke is dysregulated in the acute phase and goes up in the subacute phase [144]. This may help predict epileptogenesis after IS, but more long-term studies are needed.

The miRNA cluster 143-3p/145-3p is involved in vascular stabilization and smooth muscle contractility [145]. These miRNAs are more likely to serve as an indicator of the seizure state than of the metabolic processes associated with epileptogenesis.

miR-223 has been studied as a biomarker for epilepsy [22,41] and cerebral ischemia [84]. It is known that this miRNA affects the expression of GluR2 and NR2B and is neuroprotective [146].

miRNAs such as miR-451a are associated with the anti-apoptotic genes of BCL-2 and AKT1 proteins [147]. miR-451 also targets the drug resistance protein ABCB1/MDR1 [148], which links it to drug resistance.

miR-132 is of interest for studying the mechanisms of epilepsy. This miRNA is associated with the regulation of neuroplasticity [148,149].

Some of the above microRNAs may be of interest for the study of PTE, but more detailed studies are needed.

## 4. Conclusions

In this work, we reviewed the available information on aberrant miRNA expression in various pathological conditions. The basic problems of the available research related to miRNAs were identified. These include the timing of sampling, the type of biofluid being studied, and the individual characteristics of the diseases included in the review.

We propose a spectrum of miRNAs that should be paid attention to when studying epileptogenesis. This includes miR-21, miR-155, and miR-181a as a set of potential microRNA-based biomarkers. Also suggested is the role of miR-30b, miR-219a, miR-335, miR-342, and miR-30a. These miRNAs are not specific for epileptogenesis, but they may be of interest for further study of epileptogenic processes, both separately and in a cohort.

The prediction of epileptogenesis is of great interest to the medical community. However, the study of such processes is complex, long-term, and multifactorial. In any case, more experimental studies are needed to study miRNAs as prognostic biomarkers of epileptogenesis.

## Figures and Tables

**Figure 1 ijms-24-15366-f001:**
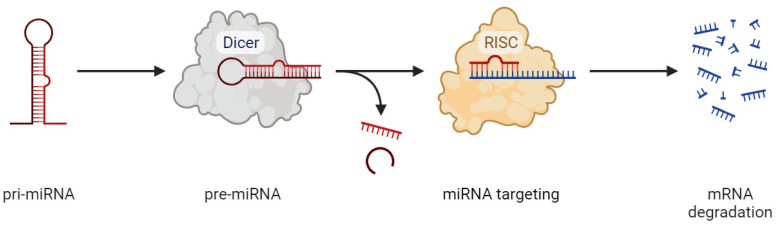
miRNA processing and mechanism of action: cytoplasmic processing of pre-microRNA via Dicer protein; disintegration of a double-stranded microRNA duplex, as a result of which only one of the strands becomes a guide; targeting guide microRNA and the RISC complex (RNA-induced silencing complex) to target mRNA; degradation of target mRNA.

**Figure 2 ijms-24-15366-f002:**
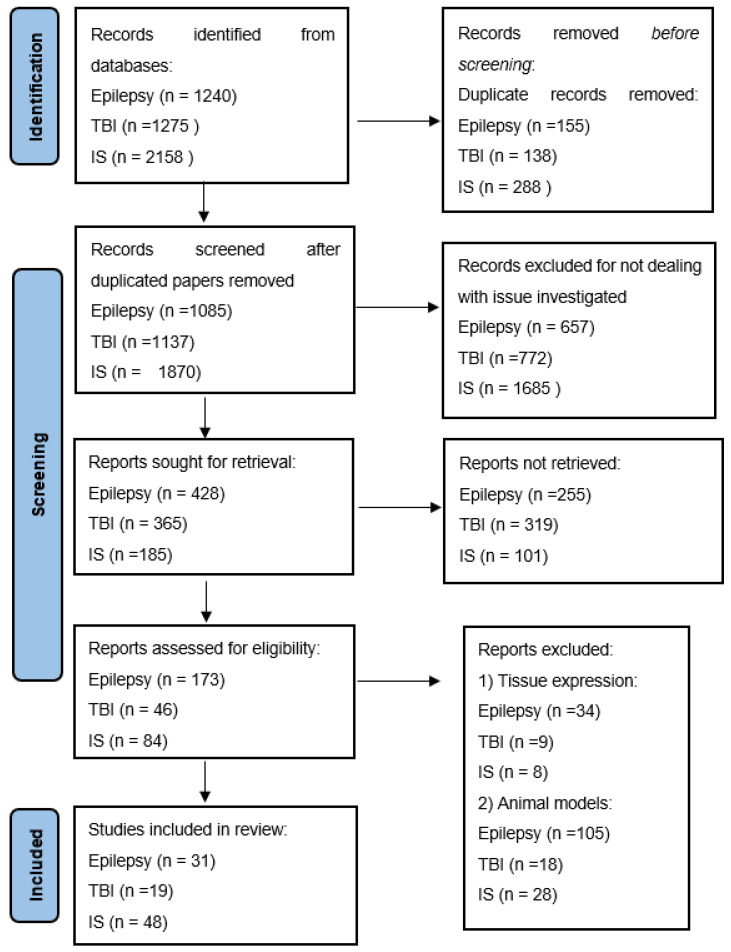
PRISMA chart.

**Figure 3 ijms-24-15366-f003:**
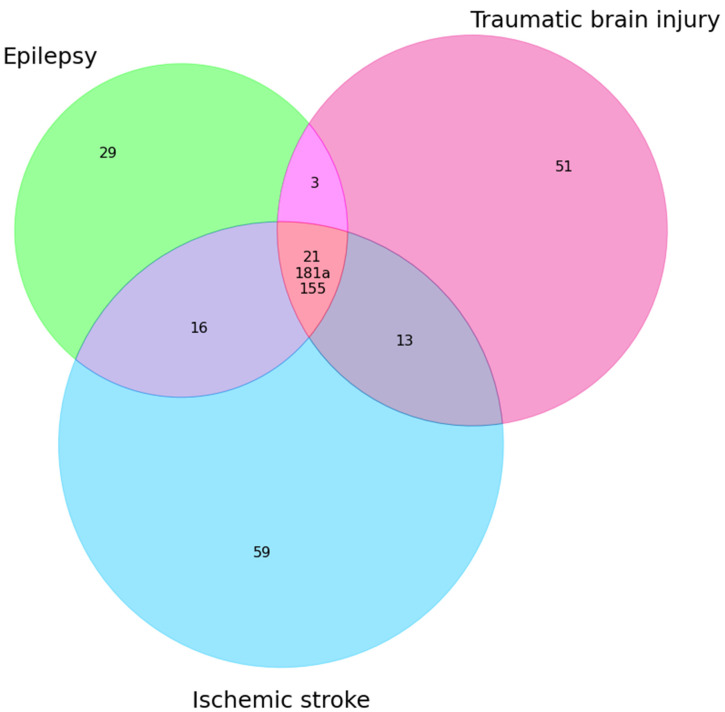
MicroRNAs common to epilepsy, TBI, and ischemic stroke. The outer circles indicate the number of miRNAs found in these diseases, at the intersection of three circles—the names of miRNAs common to all diseases.

**Figure 4 ijms-24-15366-f004:**
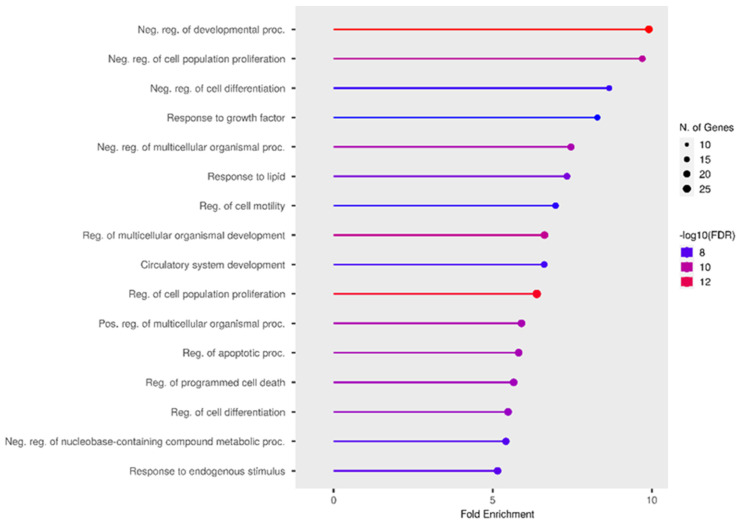
Most significant biological processes in which miR-21 is involved [22,59,82,86,100].

**Figure 5 ijms-24-15366-f005:**
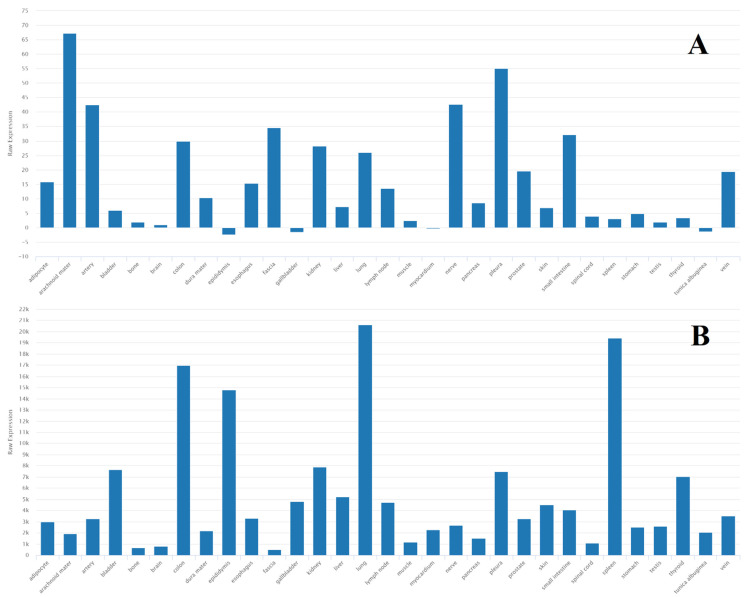
Tissue expression of (**A**) miR-21-3p and (**B**) miR-21-5p [22,59,82,86,100].

**Figure 6 ijms-24-15366-f006:**
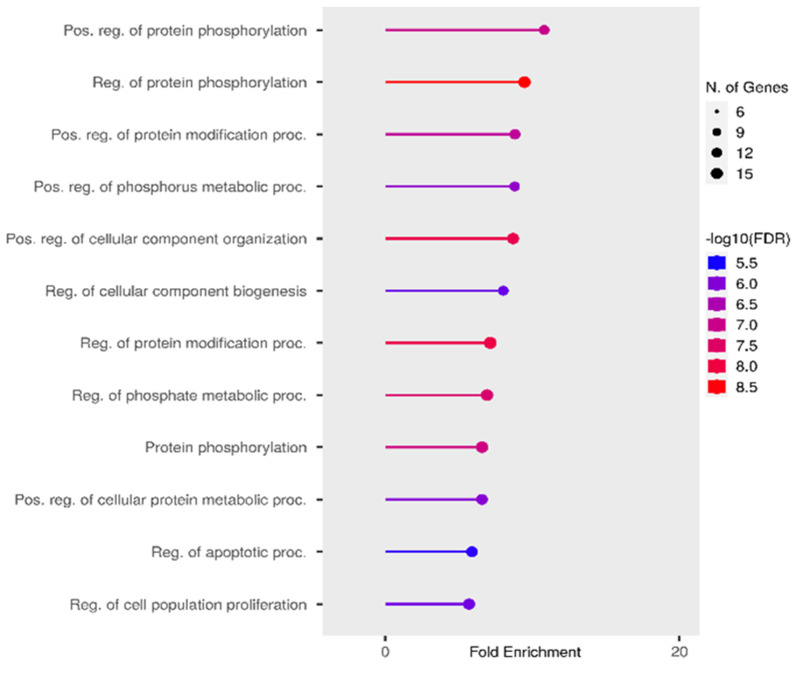
Most significant biological processes in which miR-181a is involved [39,62,100].

**Figure 7 ijms-24-15366-f007:**
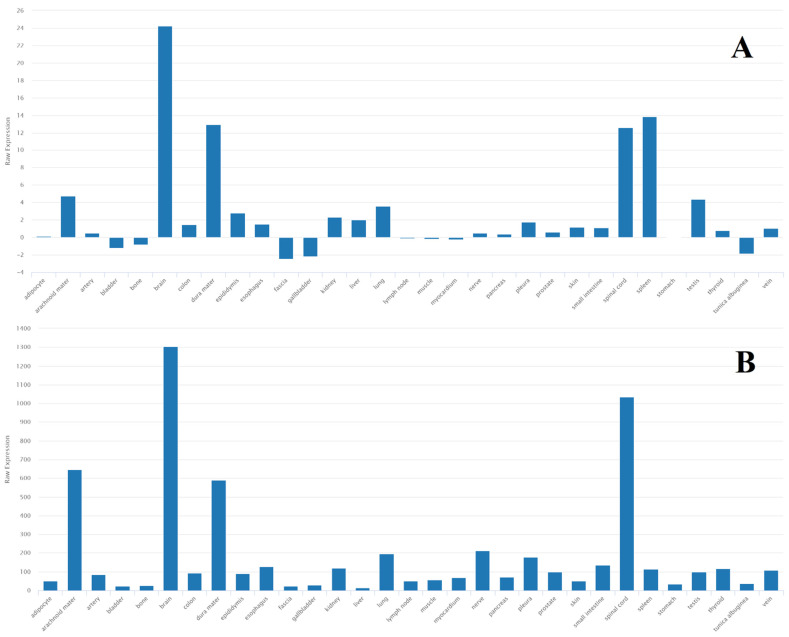
Tissue expression of (**A**) miR-181a-3p and (**B**) miR-181a-5p [39,62,100].

**Figure 8 ijms-24-15366-f008:**
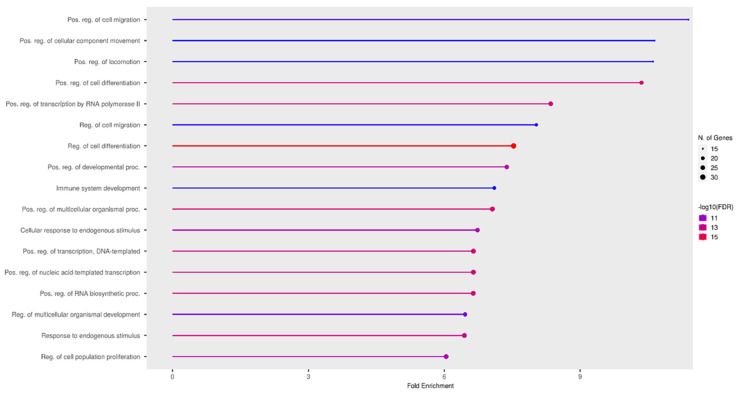
Most significant biological processes in which miR-155 is involved [27,28,29,63,115].

**Figure 9 ijms-24-15366-f009:**
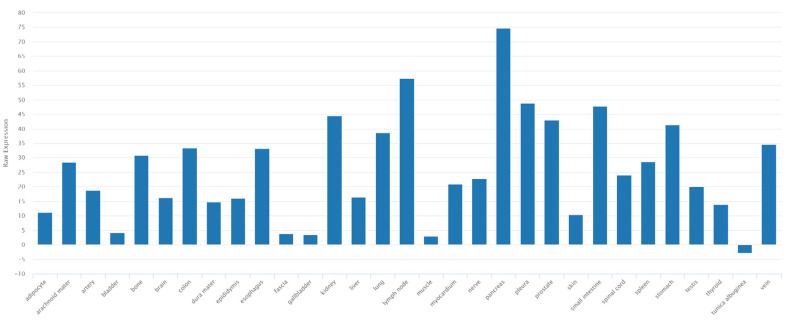
Tissue expression of miR-155-5p [27,28,29,63,115].

**Table 4 ijms-24-15366-t004:** MicroRNAs associated with epilepsy, TBI, and IS.

Epilepsy + TBI	Epilepsy + IS	Epilepsy + TBI + IS
miR-30b↑ miR-21(E↓TBI↑) miR-328-3p↑ miR-335(E↓ TBI ↑) miR-155 (E↑ TBI↓) miR-181a(E↓ TBI↑)	miR-342(E—3p,IS—5p)↑ miR-153 (E↓ IS↑) miR-106b↑ miR-146a(E↑ AIS↓) miR-30a (E↑ IS↓) miR-15a(E↑ IS↓) miR-143↑ miR-145(E↑↓ IS↓) miR-21(E↓ IS ↑↓) miR-223↑ miR-451a ↑ miR-155↑ miR-132 miR-125a(E↓ IS↑) miR-181a(E↓ IS↑) miR-142 (E↑ IS↓)	miR-21 miR-181a miR-155

E—epilepsy; TBI—traumatic brain injury; IS—ischemic stroke; AIS—acute ischemic stroke; ↑—upregulation; ↓—downregulation.

## Data Availability

Not applicable.

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
