# Peer review of "MicroRNAs as Potential Biomarkers of Post-Traumatic Epileptogenesis: A Systematic Review"

_ijms, 2023, doi:10.3390/ijms242015366_

Round 1

Reviewer 1 Report

In this study Vasilieva et al., had performed a systemic review on microRNAs as potential biomarkers in posttraumatic epilepsy. It is a well conducted study, the authors concluded that three of the candidate miRNAs could be further investigated for their potential as biomarkers.

Some minor suggestions:

1. Check all tables for language discrepancies. There is text with other language fonts. Please define for uniformity in the text legend for tables: AUC, the directionality of arrows.

2. Line 99: it is either AUC criteria or p-value. Please correct this.

3. It is not clear why some of the reports are not retrieved under screening. Please explain in the methods.

4. Line275, It is mentioned that let7b expression is different than controls. To be clear please mention weather let7b is either high or low expressed.

5. Table 3, page 13; miR-135b expression is assessed <24 hrs or 14 days after attack, or both. Please check this and clarify.

6. The fonts of the graphs of pathway analysis and tissue expression are very small.

7. The authors mention that these miRNAs could be of interest if studied as a cohort. It would be informative for future studies to mention common relevant pathways or targets that the three miRNAs are possibly regulating or targeting.

8. In Table 4, it would be good to include the stages after TBI these miRNAs were identified. Weather early, in the latency or late?

Author Response

 Dear reviewer,

Thank you very much for your comments and recommendations!

  • Thank you for drawing attention to these shortcomings. We have corrected all these errors and made the table uniform. All changes are highlighted in yellow.
  • In the studies we analyzed, the authors reported either the AUC criterion or the p value; in rare cases, they reported both. We have added a line informing about this.
  • We have added additional information to the inclusion and exclusion criteria. We also excluded studies that were not freely accessible, duplicates, and studies that were not relevant to the topic of our review.
  • We added a line about the type of altered expression for this microRNA.
  • We have added a line about the time of sampling of biomaterial for analysis of expression for this microRNA.
  • We have tried to modify the images where possible. Unfortunately, in some cases (images with tissue expression types) the figures cannot be modified without throwing out some important information.
  • Information about common gene targets and most important pathways has been added in lines 412-415.
  • This table provides general information on miRNAs that overlap between TBI and Epilepsy. Since not all studies on microRNA profiling of TBI indicated the time of sampling, we considered it inappropriate to add this information to only some of them to maintain the uniformity of the table.

Best regards, authors

Reviewer 2 Report

This paper sets out to investigate the predictive capabilities of microRNAs (miRNAs) in identifying the risk of epileptogenesis. It addresses a clinically significant issue, aiming to provide insights into a challenging problem by conducting a thorough analysis of pertinent literature. However, several areas need improvement, as highlighted below:

Introduction of miRNAs: The paper should begin with a general description of miRNAs, including their expression, production, and mechanisms. This could be effectively presented through a figure to enhance reader understanding.

Pathogenic Mechanisms and miRNAs: The paper needs to explain the pathogenic mechanisms associated with posttraumatic epilepsy (PTE), such as neuroinflammation, apoptosis, glial cell dysfunction, autophagy, and oxidative stress, and how miRNAs are related to these processes. Providing this context will strengthen the paper's foundation.

Figure 2: In Figure 2, it's crucial to clarify that the miRNA names are within the overlapping middle circles, not numbers. This clarification will prevent confusion.

Figures 3-8: Detailed explanations are necessary for these figures, including the labeling of X and Y axes and how the comparisons were made. Additionally, consider enlarging the font size throughout the paper to improve readability.

Over all, this systematic review sets out on a significant journey to investigate the potential of miRNAs as biomarkers for posttraumatic epileptogenesis. Its value is greatly enhanced by the comprehensive analysis, careful miRNA selection, and recognition of research limitations. Furthermore, the paper's emphasis on the necessity for ongoing rigorous research in this field underscores its commitment to advancing our understanding of epileptogenesis. It is imperative to highlight that addressing the raised comments concerning general miRNA descriptions, in-depth explanations of pathogenic mechanisms, figure clarity, and font size enhancement is crucial for an overall improvement in the paper's quality.

Moderate editing of the English language is needed. Please ensure that all content in tables is written in English.

Author Response

Dear reviewer,

Thank you very much for your comments and recommendations! 

All changes are highlighted in yellow.

We have added a picture briefly explaining the mechanism of action of microRNA.

We have provided brief additional information about epileptogenic mechanisms. We have tried to modify the images where possible. Unfortunately, in some cases (images with tissue expression types) the figures cannot be modified without throwing out some important information.

Regarding the axes, the images were generated using specialized browser software: ShinyGo and TisssueAtlas. The figures line up exactly this way, unfortunately we are not able to make changes to these images.

Best regards, authors